# Proteins Adsorbing onto Surface-Modified Nanoparticles: Effect of Surface Curvature, pH, and the Interplay of Polymers and Proteins Acid–Base Equilibrium

**DOI:** 10.3390/polym14040739

**Published:** 2022-02-14

**Authors:** Estefania Gonzalez Solveyra, David H. Thompson, Igal Szleifer

**Affiliations:** 1Instituto de Nanosistemas, Universidad Nacional de San Martín-CONICET, San Martín, Buenos Aires B1650, Argentina; egonzalezsolveyra@unsam.edu.ar; 2Bindley Bioscience Center, Department of Chemistry, Multi-Disciplinary Cancer Research Facility, Purdue University, West Lafayette, IN 47907, USA; davethom@purdue.edu; 3Department of Biomedical Engineering, Northwestern University, Evanston, IL 60208, USA; 4Chemistry of Life Processes Institute, Northwestern University, Evanston, IL 60208, USA; 5Department of Chemistry, Northwestern University, Evanston, IL 60208, USA

**Keywords:** protein adsorption, theoretical methods, nanoparticles, curvature, end-tethered polymers, charge regulation

## Abstract

Protein adsorption onto nanomaterials is a process of vital significance and it is commonly controlled by functionalizing their surface with polymers. The efficiency of this strategy depends on the design parameters of the nanoconstruct. Although significant amount of work has been carried out on planar surfaces modified with different types of polymers, studies investigating the role of surface curvature are not as abundant. Here, we present a comprehensive and systematic study of the protein adsorption process, analyzing the effect of curvature and morphology, the grafting of polymer mixtures, the type of monomer (neutral, acidic, basic), the proteins in solution, and the conditions of the solution. The theoretical approach we employed is based on a molecular theory that allows to explicitly consider the acid–base reactions of the amino acids in the proteins and the monomers on the surface. The calculations showed that surface curvature modulates the molecular organization in space, but key variables are the bulk pH and salt concentration (in the millimolar range). When grafting the NP with acidic or basic polymers, the surface coating could disfavor or promote adsorption, depending on the solution’s conditions. When NPs are in contact with protein mixtures in solution, a nontrivial competitive adsorption process is observed. The calculations reflect the balance between molecular organization and chemical state of polymers and proteins, and how it is modulated by the curvature of the underlying surface.

## 1. Introduction

The progress achieved in the last decades on the fabrication and manipulation of materials in the nanoscale has propelled the use of nanotechnology and nanomaterials for applications in electronics [1], development of new energy sources and environmental remediation [2], therapeutics, controlled drug delivery, and diagnostics [3]. Among the latter, the use of hybrid systems of nanomaterials that combine inorganic portions, soft matter (polymers, polyelectrolytes), and biological molecules (proteins, hydrocarbons, antibodies) appears as a promising strategy, as it conjugates the knowledge and achievements of nanotechnology, molecular biology, and medicine [4].

In this context, protein adsorption onto nanomaterials is a process of vital significance [5]. On one hand, efficient and controlled adsorption of proteins is key for their separation and purification, for their immobilization and use in biosensors and platforms for drug delivery, hyperthermia therapy, and contrast imaging. Conversely, uncontrolled protein adsorption can hinder the use of nanomaterials for biomedical applications [6]. Currently, a lot of efforts are devoted to engineering nanomaterials not only as vehicles for controlled drug delivery, but as smart nanoplatforms, responsive to their environment and with specific actions [7,8]. However, this is a difficult task, as these nanosystems combine properties of very different types of materials.

Nanoparticles (NPs) in biological environments interact with a diverse mixture of proteins, metabolites, peptides, and carbohydrates. Initially, plasma proteins adsorb onto the NPs forming a protein corona that determines the fate of the NPs [9,10,11,12]. This adsorption is nonspecific in nature, meaning it does not follow a molecular recognition interaction but rather protein–surface attractions, electrostatic and van der Waals interactions. The formation of the protein corona is a dynamic and multifactorial process that depends both on the properties of the NP (size, morphology, surface chemistry) and the properties of the medium (such as pH, ionic strength, proteins present) [13,14]. It is the complex NP–protein corona that interacts with the rest of the species in the biological system, so controlling this process is of paramount important to any biomedical application [15]. To control the nonspecific protein adsorption, the surface of the NPs are commonly passivated by fuctionalization with biocompatible polymers [16], such as poly(ethylene)glycol (PEG) [17], self-assembled monolayers (SAMS) [18,19], zwitterions [20,21], polysaccharides [22], peptoids [23,24,25], and thermoresponsive polymers [26,27]. The polymeric layer provides a repulsive steric barrier to proteins, limiting their adsorption [28,29]. The efficiency of this antifouling strategy depends on the molecular weight of the polymer and its surface density [30]. It has been found that hydrophobic or charged NPs tend to adsorb more proteins than neutral or hydrophilic ones [13,31,32].

Although significant amount of work has been carried out on planar surfaces modified with different types of polymers, studies investigating the role of surface curvature are not as abundant [13,32,33,34]. Protein adsorption onto gold NPs (Au-NPs) of different sizes modified with PEG molecules was found to decrease as the surface density of polymer increases and the NPs size increases [35]. The conformational freedom of the chains depends on the curvature of the NP and the surface density of the polymers, and so it affects the entropic barrier they present to proteins [31,36]. This relation between NP curvature and polymer density determines not only the total amount of adsorbed proteins but also the composition of the protein corona. Moreover, in curved surfaces, even modifying the NPs with high surface density of PEG molecules does not achieve a complete elimination of protein adsorption. This actually fuels the research and development of biocompatible polymers with improved antifouling properties [16].

The interplay between NP morphology and curvature and the chemistry of the polymers on its surface has been phenomenologically studied, but there there is still a need for a comprehensive understanding of the physicochemistry in place. The antifouling properties of a large library of Au-NPs of different sizes and types of ligands on their surface was obtained by resorting to combinatorial strategies and bioinformatics tools [33]. It was found that proteins adsorb more onto NPs modified with charged molecules (either positively or negatively charged) than NPs of the same size with neutral coatings. Comparing NPs with the same type of surface chemistry, small NPs adsorb higher density of proteins, and the magnitude of this effect depends in turn on the nature of the surface coating (neutral, anionic, or cationic). These results highlight the nontrivial interplay between the NP’s curvature and its surface chemistry over the protein–protein and protein–NPs interactions.

The cited results and references show that there is a large number of engineered nanosystems with antifouling properties. However, the experimental information is disperse and heterogeneous [11,37,38]. Recent efforts combine NP libraries with the development of nanodescriptors to find correlations that would accelerate the discovery of NPs with tunable properties [39,40,41]. Although these strategies are useful for the prediction and the design of engineered NP, they do not provide a fundamental understanding of the physical chemistry of the systems, a knowledge that could lead to expanding their scope of applicability. In this way, it is crucial to achieve a comprehensive description on how the antifouling properties depend on the design parameters of the modified NPs and how they shape the nano–bio interface [42,43]. This is a challenging task given the complexity of these multicomponent systems, with hierarchical and multiscale interactions that couple with structure, molecular organization, and chemical reactions [44]. Common experimental techniques provide very useful information on the adsorption process, but this data is statistically averaged, with limited insight on the interactions at the molecular level [5]. Moreover, it is also difficult to control and systematically vary experimental parameters such as pH and ionic strength. This provides an opportunity for adequate schemes of theoretical modeling to contribute and expand the progress in the field of protein adsorption [44,45,46,47].

Most published results on protein adsorption onto surfaces and NPs derived with simulation methods are mainly focused (although not exclusively) on inorganic particles or carbon-based systems, without any surface modification with polymers [48,49]. Our goal in this work is to gain a fundamental understanding on the interplay between the acid–base equilibrium of polymers and proteins as they adsorb onto end-tethered polymers of different nature and the curvature of the underlying grafting surface, thus advancing on previous theoretical studies. Our working hypothesis is that the conditions of the solution (pH and ionic strength), the details of the engineered surface (type of polymers, their surface density), and the coupled charge regulation between proteins and grafted polymers (when acidic or basic), modulated by the surface curvature, determine the protein adsorption process. To that end, we performed molecular theory calculations on surfaces of different morphology and curvature, modified with a mixture of polymers of different lengths and chemical properties, consisting of short neutral chains and long ones that can be either neutral, acidic, or basic. The reason for using polymer mixtures derives from the interest of having short chains that prevent protein adsorption and long ones that can be further functionalized for other applications, such as molecular recognition. We studied the modified NPs in contact with solutions of different pH values containing different concentrations of sodium chloride and different protein concentrations. Lysozyme and green fluorescent protein (GFP) were chosen for this study. Both proteins are water-soluble but have very different sizes, shapes, and isoelectric points, and in this way they allow studying these parameters in the adsorption process [5,50]. Figure 1 summarizes the modeled systems.

The paper is organized as follows: in the next section we provide a description of the theoretical approach and molecular models for proteins and polymers employed in our calculations. We start presenting the results for protein adsorption from one protein solution onto NPs of different curvature and morphology modified with neutral coatings. Analysis of the effects of the solution’s pH and salt concentration in these neutral NPs is followed by the study of the interplay between the acid–base equilibrium of polymers and proteins for NPs functionalized with weak polyelectrolytes. Competitive adsorption from binary protein mixtures is then discussed. Finally, we conclude by presenting the main points of this work, tying together the key factors that govern the process of protein adsorption onto modified NPs, and by proposing highlights for future lines of work.

## 2. Materials and Methods

### 2.1. Theoretical Approach

To study the thermodynamics of protein adsorption onto modified NPs we employed a theoretical approach based on a molecular theory that takes into account the size, shape, conformation, and charge of each molecular species in the system [30,51,52]. Very importantly, it allows to explicitly consider the acid–base reactions of the titrable amino acids in the protein and the monomers in the grafted polymers [53]. The general methodology has been successfully implemented to describe systems with acid–base reactions [54,55], ligand–receptor binding [56,57], redox reactions [58], and ion pair formation [59,60]. Relevant for this work, protein adsorption onto planar surfaces modified with peptoids were in good agreement with experimental observations [23,61].

The key idea of the molecular theory is to express the free energy of the system as a function of the probabilities of each macromolecule conformation (polymers and proteins), the spatial distributions of mobile species (anions, cations, solvent molecules), the electrostatic potential, and the fraction of charged species (either titrable monomers or amino acids). The Helmholtz free energy of the system has the following contributions:(1)$$F=-TS_{conf,pol}-TS_{mix}-TS_{TR,prot}+E_{ads,prot}+F_{chem}+E_{elect},$$
where *T* is the temperature; $S_{conf,pol}$ is the conformational entropy of the tethered polymers chains; $S_{mix}$ corresponds to the mixing (translational) entropy of the small mobile species (water molecules, anions, cations); $S_{TR,prot}$ is the translational and rotational entropy of protein molecules; $E_{ads,prot}$ correspond to the adsorption energy of the proteins; $F_{chem}$ represents the free energy associated with protonation and deprotonation reactions; and $E_{elect}$ is the total electrostatic energy functional. Each of the free energy terms is written as a functional of the density distribution of their molecular components and the probability distribution function of the polymer/proteins conformations. The system is assumed to be incompressible and the steric repulsions between all molecular species are included as a constraint to the free energy. The distribution profiles, the probability of the different conformations, the electrostatic and repulsive position-dependent potentials, and the chemical state (protonated/deprotonated) of each titrable species are determined through the minimization of the total free energy. A comprehensive description of the theoretical framework can be found in [62,63] and in the Appendix A.

The input to the theory includes all the parameters of the systems: a very large set of unbiased polymer conformations, the surface density of the end-tethered polymers and the composition of the surface mixture (fraction of short and long polymers), a set of protein conformations, the protein adsorption energy, the volume and charge of all molecular species, the equilibrium constants (pK) of all acid–base chemical reactions, the bulk solution conditions (pH, salt, and protein concentrations), and the radius and geometry of the surface (spherical, cylindrical, or planar).

Minimizing the free energy provides the amount of adsorbed protein (adsorption isotherms), the local concentrations of mobile species, the probability of every polymer conformation, the charge state (protonated/deprotonated) of every titrable species, and the local electrostatic potential, among other thermodynamic and structural properties of the system [53].

### 2.2. Molecular Models

In order to solve the theory, we need a set of chain conformations for each type of polymer (short and long). In principle, these sets should include all allowed conformations that do not collide with the surface for the geometry under study, but in practice it is enough to use a very large set of randomly chosen conformations, which we generate in free space using the three-state rotational isomeric state (RIS) model [64]. To avoid biases, each random bond sequence is rotated using randomly chosen Euler angles. Only self-avoiding conformations that do not overlap with the surface are considered. We have used a set of 10${}^{6}$ different configurations for each polymer type and each geometry. Further details on the polymer model can be found in Appendix A.

Proteins are modeled with a coarse grain model where each amino acid is represented by a single solid bead centered at the position of the corresponding α-carbon (Appendix A) [62,63]. The position and sequence of all atoms in the proteins are taken from the crystallographic structure PDB files (193L [65] and 1EMA [66] for lyzozyme and GFP, respectively). Lysozyme is known to undergo negligible conformational changes upon adsorption [5,67], while the β-barrel of GFP is known to be stable and rigid [68]. Hence, in our model, we do not take into account conformational changes upon protein adsorption on the NP surface. The relative position of all beads remains frozen to the initial structure of the PDB structure, irrespective of solution conditions. In this way, the proteins are modeled as rigid bodies, while retaining full translational and rotational degrees of freedom. The volume of each coarse grain bead is taken as the molecular volume of the amino acid, which was computed using the package VOIDOO [69] (see Appendix A). Amino acids are considered hydrophilic and are classified either as neutral or titrable. Among the latter, aspartic acid (ASP), glutamic acid (GLU), and tyrosine (TYR) are considered acidic groups, while arginine (ARG), histidine (HIS), and lysine (LYS) are basic. Each titrable bead is characterized by an intrinsic acidic constant, while all other amino acids are considered neutral. The pKa values for the titrable amino acids correspond to experimental values averaged over different proteins [70]. Further details on the protein models can be found in Appendix A.

## 3. Results and Discussions

In the following sections, we present and discuss the nonspecific protein adsorption process onto surfaces of different morphology and curvature, modified with a mixture of polymers of different length and monomer type (neutral, acidic, basic). We study the modified NP in contact with solutions of different pH values containing different concentrations of sodium chloride and different protein molecules. To characterize protein adsorption, we quantify the protein surface excess:(2)$$\Gamma_{prot}=\int_{R}^{\infty }\left[\langle \rho_{prot}\left(r\right)\rangle -\rho_{prot}^{bulk}\right]G\left(r\right)dr$$
where $\rho_{prot}\left(r\right)$ refers to the local concentration of prot∈ {lysozyme, GFP} at position *r* and $\rho_{prot}^{bulk}$ to its bulk concentration. The function G(r) = A(r)/A(R) describes the change in volume as a function of the distance away from the tethering surface [36].

The results presented correspond to a set value of adsorption energy, $\epsilon_{ads,prot}=-50k_{B}T$ (124 KJ/mol for T = 298 K), which is in the range of previously used values [30]. However, to complete the analysis we also performed calculations varying this parameter, as we will discuss and analyze further below.

### 3.1. Curvature Effects on Protein Adsorption

We start discussing the main features of the protein adsorption process by studying the adsorption from single-protein solutions of lysozyme onto planar, cylindrical, and spherical surfaces modified with a mixture of neutral polymers. Although there is abundant amount of literature for lysozyme adsorption onto modified planar surfaces, both experimental and theoretical, the adsorption onto curved NPs is less characterized.

Surface curvature modulates the adsorption process and the molecular organization in space. Figure 2 shows the density of adsorbed protein as a function of the total surface density of polymers, for a 1:1 mixture of long and short PEG polymers grafted on spherical NPs of different sizes, for a given bulk solution salt concentration and two illustrative pH values (11.0 and 5.0). The corresponding figure for cylindrical NPs can be found in the Appendix A. Irrespective of the curvature of the surface and the pH of the bulk solution, we can see that increasing the amount of total polymers on the surface hinders protein adsorption, given the ability of the polymer layer to present a steric barrier to the proteins in solution. In the case of neutral polymers, the amount of adsorbed lysozyme results from the balance between the bare surface–protein attraction, the protein–protein and protein–polymer steric repulsions, the protein–protein electrostatic repulsions, and the loss of conformational entropy of the polymer chains, since they need to stretch away from the surface in order to accommodate the adsorbing proteins [51]. Increasing grafting density increases the crowding on the surface and with that the steric repulsions within the interfacial region, decreasing protein adsorption and even suppressing it for high enough values. The same behavior is observed when increasing the ratio of long polymers in the surface mixture for a fixed grafting density (Appendix A), following the analysis just provided. It is worth mentioning that our calculations show monolayer protein adsorption. This follows from the fact that in our current model, protein–protein interactions are mainly repulsive (both electrostatic and steric), since no van der Waals interactions were included. Lacking attractive protein–protein interactions hinders multilayer adsorption processes. Including these interactions is beyond the scope of the current work, but will be subject of futures studied, as discussed in the Conclusions section.

Regarding the effect of the surface curvature, we observe that lysozyme adsorption decreases as the size of the NP increases. In turn, adsorption on spherical NPs is larger than on cylindrical ones (see comparison for pH = 11.0 in Appendix A). This is due to the fact that in curved convex surfaces, the available volume increases as we move away from the surface, scaling as r/R for cylinders and as (r/R)${}^{2}$ for spheres (where r is the radial direction and R the radius of the surface). Increasing the size of the NP decreases the available volume at a given distance from the surface, thus increasing the protein–protein and protein–polymer steric repulsions within the layer. This ultimately leads to a decrease in protein adsorption. For the limiting case of infinite radius, we retrieve the results for adsorption onto a planar surface, which we can see as a lower bound. This is also reflected on the molecular organization in interfacial region. Appendix A depicts the volume fraction profiles of the short and long PEG polymers as well as the protein volume fraction. As before, given that the available space as we move away from a planar surface is smaller than for a curved NP, the polymer layer end-grafted to a planar surface is more stretched towards the solution and it poses a better steric barrier to prevent protein adsorption. On the other hand, for a spherical NP, the volume change is larger, the polymers are more dispersed, and the steric barrier is not as effective. Proteins can move closer to the surface and adsorb more easily.

Our calculations show that, for a given pH value, there is an effect of surface curvature and morphology on the adsorption process, but quantitative, rather than qualitative. However, when comparing isotherms for bulk pH values of 11.0 and 5.0 (Figure 2 right and left panels, respectively), we see an important effect. Proteins are amphoteric molecules that contain both acidic and basic groups, for which degree of charge depends critically on the pH and ionic strength of the bulk solution. These are the determining factors of the adsorption process, as we discuss further next.

#### Effect of Solution Conditions: pH and Salt Effects

Figure 3 shows the adsorption of lysozyme as a function of the bulk solution pH and salt concentration. Comparing the behavior on planar, cylindrical, and spherical surfaces, we can see that the above discussion still holds at different solution conditions, though curvature effect is significantly smaller than that of the bulk pH or ionic strength in the millimolar range. This highlights that even though surface curvature modulates adsorption, the dominant interactions in the systems are electrostatic in nature.

Adsorption onto neutral polymer-coated surfaces depends nonmonotonically on the bulk solution pH, which can be rationalized considering its effect on the net charge of the protein when going from very acidic to very basic pH (see also Figure 4 below). At low pH values, the protein bears a high positive charge: protonated acidic amino acids are neutral, while protonated basic ones are positively charged. Increasing the pH decreases the degree of charge of basic residues, while increasing that of acidic groups, leading to an overall negative charge at very high pH values. With that in mind, adsorption increases as the net charge of the protein decreases, and it is maximum when the bulk solution is similar to the isoelectric point of the protein (for lysozyme, the calculated isoelectric point is pI = 10.99, in agreement with experimental values [71]). Regarding the effect of ionic strength, increasing salt concentration leads to an increase in the amount of adsorbed lysozyme for the entire pH range. However, the magnitude of this effect depends on the pH. Salt ions in the solution act by screening the charges in the system. For the case of proteins interacting with a surface modified with neutral polymers, electrostatic interactions are limited to the protein–protein repulsions (for titrable polymer layers this is not the case, as we will discuss below). Hence, in these systems, increasing ionic strength reduces electrostatic repulsions which translates to greater protein adsorption. As can be seen in Figure 3, this is more marked for low pH values, for which lysozyme is positively charged, and protein–protein electrostatic repulsions play a significant role in the overall interplay that governs protein adsorption. Modifying the parameters of the polymer surface mixture (that is, the total grafting density of the polymers and the fraction of long polymers in the mix) does not affect the underlying nature of that interplay, irrespective of pH, as can be seen in Appendix A. Given the neutral nature of the monomers, increasing the amount of monomers on the surface (either by increasing the total grafting density or the ratio of longer polymers) increases the steric repulsions in the layer, hindering protein adsorption. This results in a monotonic decrease of the adsorbed amount of lysozyme onto the surface, irrespective of pH, salt concentration, or surface curvature.

The above analysis encouraged us to study the interplay between the acid–base equilibrium of the amino acids in the proteins and that of the polymers grafted to the NP surface. Therefore, next, we analyze in detail the effect of changing the type of segment of the long polymers in the surface mixture, considering them neutral, acidic, or basic.

### 3.2. Interplay of Polymers and Proteins Acid–Base Equilibrium

Now, we turn our attention to surfaces modified with a polymer mixture consisting of short neutral polymers and long polymers whose segments can either be neutral, acidic, or basic (with pKa values of 5.0 and 9.0, respectively). We find that for convex NPs modified with acidic or basic polymers, surface curvature does not critically alter the protein adsorption process, or the forces governing it, as shown in Appendix A. Hence, going forward, we focus our analysis on cylindrical NPs of radius R = 5 nm. Unless otherwise stated, we fixed the salt concentration value to 1 mM, since it allows enhancing the effect of electrostatic interactions in the systems. However, to complete the analysis, we also performed calculations varying this parameter, as we will discuss and analyze further below.

Figure 4 shows the effect of polymer acid–base behavior on the adsorption of single-protein solutions of lysozyme and GFP. The choice of proteins allows analyzing the impact of the protein pI, relative to the polymer pKa, on the amount of adsorbed protein. In terms of general features, the dependence of Γ on the solution pH is nonmonotonic for all polymer types, as discussed above for the case of neutral polymers. However, changing the nature of the long polymer dramatically enhances or hinders the adsorption process. In doing so, we are changing the protein–polymer electrostatic interactions in the system (adding to the excluded volume, the protein–protein electrostatic repulsions, and the protein–surface interactions mentioned above), which can be attractive or repulsive, depending on the bulk pH and the relative value of the protein pI and the polymer pKa. The balance of forces in these scenarios is very sensitive to the protein–polymer acid–base behavior pair, the pH, and the salt concentration of the bulk solution. This means that the same polymer coating can act by preventing or promoting protein adsorption, depending on the conditions of the solution, which makes these smart systems sensitive and responsive to the environment. Our results on curved NPs are in line with previous analysis of lysozyme adsorption onto planar surfaces modified with pH-responsive hydrogels [62,63,72].

Starting with single-protein solutions of lyzozyme, adsorption is greatly enhanced when the polymer is acidic, as compared to both neutral and basic and to the bare surface (Figure 4, left panel). At low pH values, the protein has a high positive charge, while the polymer is slightly charged (see also the average fraction of charged monomers in Appendix A). Still, this is enough to significantly increase the amount of adsorbed protein. When increasing pH, as the polymer pKa is approached, the fraction of negatively charged monomers increases. Even though there is a decrease in the total (positive) charge of the protein, this leads to strong protein–polymer electrostatic attractions that translate into increased adsorbed protein. In this line, experimentally, lysozyme was also found to readily adsorb onto silica NPs negatively charged at physiological pH (∼7.4), even when modified with neutral or zwitterionic polymers [32]. Although our conditions and the experimental conditions in the previous reference are not identical, the results serve to qualitatively support our calculations. Increasing pH even more and approaching the protein pI, the acidic polymers are completely deprotonated and negatively charged, but the protein total charge decreases, along with the electrostatic attractive forces between them, and so protein adsorption starts to go down. At sufficiently high pH, the net charge of the lysozyme is also negative, leading to strong repulsions with the negatively charged polymer layer that result in a sharp drop of $\Gamma_{lys}$.

For the case of NPs modified with basic polymers, we see that the coating acts as antifouling, that is, it decreases protein adsorption with respect to the bare surface (as opposed to the acidic case). Given that pI (10.99) > pKa (9.0), there is no pH range for which polymers and proteins are oppositely charged. The resulting attractive forces would oppose the repulsive forces of steric interactions (as discussed for the acidic layer). For pH values above the polymer pKa, the layer is mostly neutral and we see that the system behaves as it does with a neutral polymer coating. However, decreasing pH leads to a protonation of the basic monomers, and this positive charge results in strong electrostatic repulsion with the also positively charged protein, leading to negligible protein adsorption.

Analyzing now the case of single-protein solutions of GFP, we see the importance of the relative values of the polymer pKa and the protein pI and also the protein size (or volume). Regarding protein size, we observe that, in general, irrespective of the type of NP coating, adsorption of GFP is lower than for lysozyme (note that the plots corresponding to $\Gamma_{GFP}$ and $\Gamma_{lys}$ are in the same scale). This can be rationalized comparing the size of lysozyme (129 amino acids) to that of GFP (238 amino acids). This implies that steric repulsions in the adsorption of GFP are much more significant. Turning our attention to the polymer coating, we observe that the fact that a given coating prevents or promotes protein adsorption depends not only on the acid–base behavior of the polymer, but also on the pH of the solution for that same coating. As before, this can be rationalized taking into account the net charge of the adsorbing protein and how it changes with pH (Figure 4, left panel). For acidic polymers at pH values close to the monomer pKa, since this value is lower than the GFP pI (6.33), the polymer layer is negatively charged (see also the average fraction of charged monomers in Appendix A), while the protein is positively charged, increasing the amount of adsorbed proteins. However, increasing the pH further leads to electrostatic repulsions, rapidly decreasing $\Gamma_{GFP}$. Comparing with the behavior on a bare surface, we observe that the polymer layer promotes adsorption for pH values in the range of 5.0–6.33 (roughly corresponding to the monomer pKa and the protein pI, respectively), while it completely prevents it for pH > 8.0. Qualitative similar results were observed experimentally for the adsorption of β-lactoglobulin, a protein of similar isoelectric point, onto negatively charged silica NPs [34]. Switching now to the case of NPs coated with basic polymers, we observe that the difference between pKa and pI makes much more possible than in the lysozyme case, particularly given that pI${}_{GFP}$ < pKa (9.0). For high pH values, for which monomers are deprotonated and neutral, there is not much of a difference in the adsorption of GFP onto NPs with neutral coatings. However, as the pH decreases, the fraction of protonated monomers increases, resulting in a positively charged layer (Appendix A). For the region between the monomer pKa and the protein pI, this results in attractions with the negatively charged adsorbing protein. Once the pH drops below the protein pI, adsorption starts to decrease, given the electrostatic repulsions. Again, we observe that, with respect to the bare surface, a basic coating con be antifouling or protein-adsorbing depending on the conditions of the solution.

It is interesting to note that when NPs are grafted with titrable monomers (either acidic or basic), the impact of surface details on protein adsorption, such as polymer surface density ($\sigma_{tot}$) or composition of the polymer mixture (x${}_{l}^{surf}$), may be quite different to what is observed for neutral coatings. Figure 5 collects the results of lysozyme adsorption from single-protein solutions onto polymer coatings of different nature, as a function of pH for different grafting densities ($\sigma_{tot}$).

As discussed in the previous section, increasing $\sigma_{tot}$ for neutral polymers leads to a monotonic decrease in $\Gamma_{lys}$ (as shown in Appendix A). In those systems, changing the amount of monomers on the surface only alters the excluded volume interactions between the species in the interfacial region. However, when the monomers are titrable, electrostatic interactions also come into play, governed by the pH and salt concentration of the solution. For the acidic coating, we observe that the dependence of $\Gamma_{lys}$ with $\sigma_{tot}$ may be of monotonic increase (for pKa < pH < pI${}_{lys}$) or monotonic decrease (pH < pKa and pH > pI${}_{lys}$), depending on the pH. The same applies for the impact of x${}_{l}^{surf}$ on $\Gamma_{lys}$, since increasing its value means increasing the fraction of titrable long polymers in the polymer mixture, as shown in Appendix A. Meanwhile, when the NP is coated with basic polymers, given that pI${}_{lys}$ > pKa, there is no pH range for which electrostatic attractions are in place. The layer behaves very similar to a neutral one for pH > pKa, for which the monomers are deprotonated and neutral, with excluded volume repulsions being dominant. For pH < pKa, electrostatic repulsions move in the same direction as the steric repulsions, such that increasing the amount of basic monomers at the surface decreases the amount of adsorbed protein even further. Thus, for the basic coating, increasing $\sigma_{tot}$ or x${}_{l}$ leads to a monotonic decrease in $\Gamma_{lys}$ (Appendix A). A similar physicochemical rationale can be made to understand the impact of surface parameters on the adsorption of GFP from single-protein solutions, although the results are very different (Appendix A). The relevant parameters are the same, although one must take into account how the pKa of the polymer (acidic or basic) compares to the pI of the GFP.

It is very interesting to analyze what happens to the local pH as a consequence of the adsorption of proteins. It should be clarified that in the context of our work, the expression *local pH* is short for the negative common logarithm of the local proton concentration, and it does not correspond to the local value of pH, to avoid conflicts with the IUPAC definition of pH [73]. Figure 6 collects the local pH as a function of the distance to a cylindrical NP in contact with lysozyme solutions at different pH values. The NP is modified with different types of polymer mixtures (neutral, acidic, basic), as indicated in the figure. Note that the amount of adsorbed protein is different for each bulk pH (see Figure 4).

For the neutral polymer system (left panel of Figure 6), we observe that when there are no proteins in solution (dotted lines), the local pH is that of the bulk except at very short distances from the surface, where a slight increase is observed. This is due to the entropic cost associated with confining protons inside the neutral polymer layer. There are less protons, hence the local pH increases. Now, when there are proteins in solution that adsorb onto the surface, the local pH in the vicinity of the NP can differ from the bulk pH up to one pH unit. The local pH increases drastically close to the surface when the pH is smaller than the pI of the protein. At these bulk pH values, the adsorbing proteins are positively charged, and repulsions with protons near the surface lead to an increase in the local pH. For pH values higher than the protein pI, the local pH in the vicinity of the NP is actually lower than that in the bulk. In these conditions, the protein has a negative net charge, which attracts protons inside the polymer layer, leading to a decrease in local pH. The results shown in Figure 6 correspond to single-protein solutions of lysozyme, but the analysis is analog to solutions of GFP, as shown in Appendix A.

For the NP modified with acidic or basic polymers (central and right panels of Figure 6), the local pH in the interfacial region differs greatly from that of the bulk even when no proteins are present in solution, in line with recent coarse-grained simulations and experiments of weak polyelectrolytes in solution [74,75], and grafted weak polyelectrolytes on curved NPs [54]. For the acidic coating, the deprotonation of the monomer units leads to negative charges within the layer that attract protons and decrease the local pH (no proteins in solution) [54]. Now, when adding lysozyme to the solution, at pH < pI, the adsorbing proteins have a positive charge, decreasing the amount of protons in the interfacial layer and raising the local pH. This effect is relevant even for very low pH values, for which the amount of adsorbed lysozyme is quite small (see also Figure 4 at pH = 3.0). Increasing the pH of the solution leads to further deprotonation of acidic monomers and also to a decrease in the positive charge of adsorbing lysozyme (Figure 4 left panel), so that the local pH in the vicinity of the surface decreases as a consequence of more protons in that region to compensate negative charges.

For NPs modified with basic polymers, calculations for pH values below the pKa (9.0) show that the protonation of the monomers leads to positive charges in the layer that repel protons from the interfacial region, increasing local pH dramatically (even with no proteins present). The effects of lysozyme adsorption on the local pH become significant only in the pH region of its pI, since below that value, proteins are also positively charged. Once the adsorbing proteins have a negative net charge (pH > pI${}_{lys}$), we see a decrease in local pH, a consequence of attraction of protons inside the layer.

The changes in the local concentration of charges discussed thus far translate into changes in the electrostatic potential, and this has large effects on the acid–base equilibrium of the titrable amino acids of the adsorbing proteins as they approach the surface, as has been computed for planar systems [62]. To further explore this response to the local environment in curved NPs, we compared how their acid–base behavior differs in the interfacial region upon adsorption as compared to the protein in bulk solution, and how it acts with the acid–base behavior of the polymers on the surface mixture. To that end, we computed the average degree of charge for each titrable residue as the weighted average of the spacial degree of (de)protonation:(3)$$\langle f_{aa}\rangle =\frac{\int_{R}^{\langle H_{pol}\rangle }f_{aa}\left(r\right)\langle \rho_{aa}\left(r\right)\rangle G\left(r\right)dr}{\int_{R}^{\langle H_{pol}\rangle }\langle \rho_{aa}\left(r\right)\rangle G\left(r\right)dr}.$$
where $f_{aa}$ and $\langle \rho_{aa}\left(r\right)\rangle$ are the local degree of charge and density of the titrable amino acid, respectively. The function G(r) describes the change in volume as a function of the distance away from the tethering surface [54]. $\langle H_{pol}\rangle$ is the height of the polymer layer, which corresponds to twice the first moment of the normalized r-dependent density of the monomers, minus the radius *R* of the nanoparticle:(4)$$\langle H_{pol}\rangle =2\left(\langle r\rangle -R\right)\;with\;\langle r\rangle =\frac{\int_{R}^{\infty }r\langle \rho_{p}\left(r\right)\rangle G\left(r\right)dr}{\int_{R}^{\infty }\langle \rho_{p}\left(r\right)\rangle G\left(r\right)dr}.$$

The height, as defined above, is a measure of the extent of the polymer layer. Figure 7 collects the results for NPs coated with neutral, acidic, or basic polymers in contact with solutions containing no proteins, single-protein solutions of lysozyme or GFP, or binary protein mixtures in solution. Note the important difference in polymer height between neutral polymers and weak polyelectrolytes. This is due to electrostatic repulsions arising in the latter for acidic or basic monomers as we increase or decrease the pH, respectively, since this translates into more same-charged monomers. For solutions containing lysozyme, there is also an increase in the length of the polymer layer for pH ∼ pI as a result of steric repulsions derived from the adsorbed proteins within the layer.

Figure 8 shows the average fraction of deprotonated (acidic, upper panels) or protonated (basic, lower panels) amino acids of adsorbed lysozyme onto cylindrical NPs (R = 5 nm) as a function of bulk pH. The values corresponding to the amino acids in proteins in bulk solution are also included for comparison (results for adsorbed GFP can be found in Appendix A).

We can see that the acid–base behavior of the polymers has a great impact on that of the titrable amino acids, especially when the pKa values of the amino acids and the monomers are similar. This nontrivial amino acid (de)protonation has also been characterized in lysozyme adsorbing onto planar surfaces modified with pH-responsive hydrogels [62,72]. For each bulk pH value, the system will minimize the free energy of the system via a complex balance between the protein–protein and protein–polymer steric repulsions, the protein–protein and protein–polymer electrostatic interactions, and the protein–bare surface attractions, given the salt concentration, the surface details ($\sigma_{tot}$ and x${}_{l}^{surf}$), and the acid–base behavior of the polymer coating.

Let us start analyzing systems with neutral coatings. In them, there is no interplay between charging of titrable monomers and amino acids. Steric repulsions and electrostatic repulsions of same-charge proteins oppose the attraction between the bare surface and the proteins. In that scenario, reducing the overall net charge of the protein will allow minimizing the total system free energy. For pH values < pI${}_{lys}$ (for which the adsorbing proteins are positively charged), we observe an upregulation of acidic monomers as compared to amino acids in bulk protein solutions (Figure 8 upper panels, blue lines). This means that the average fraction of deprotonated and negatively charged amino acids is bigger for the same pH value. At the same time, the basic amino acids are downregulated (Figure 8 lower panels, blue lines). Now, for pH values > pI${}_{lys}$ (for which the adsorbing proteins are negatively charged), the opposite is observed: upregulation of basic amino acids and downregulation of the acidic ones. The overall goal with this strategy is to reduce the net charge of the adsorbed proteins at each given pH, in order to reduce electrostatic repulsions (see the amount of each type of amino acid in the protein in Appendix A).

When the NPs are coated with acidic or basic polymers, charge regulation of the amino acids is going to be coupled with that of the titrable monomers, in order to maximize attractions and minimize repulsions. For the acidic amino acids, we can see that, in general, an acidic coating on the surface decreases their average degree of charge, as compared to amino acids in bulk protein solutions for the same pH value. This reduces electrostatic repulsions between the deprotonated amino acids and the monomers on the surface but also increases attractions between the overall positively charged protein and the deprotonated monomers. It is worth noting though, that this downregulation of amino acid charge is reversed for very low pH values for aspartic acid and glutamic acid. At those pH values, the polymers are protonated and neutral (Appendix A). Now the system will try to minimize protein–protein electrostatic repulsions by increasing the negative charge on acidic amino acids with pKa values in the acidic range. This is the same as discussed above for neutral layers. For the basic amino acids, an acidic coating on the surface increases their average degree of charge, as compared to amino acids in bulk protein solutions for the same pH value. In this way, the overall positive charge of the protein increases, and with that the electrostatic attractions with the acidic polymer layer.

Finally, when the coating of the NP is basic, a significant upregulation of the acidic amino acids is observed for pH < pI${}_{lys}$. At those pH values, the adsorbing proteins are positively charged, which can interact with intense electrostatic repulsions with the also positively charged protonated monomers at the surface. For tyrosine, we observe that this upregulation is reversed for pH ≈ pI, given that the adsorbing lysozyme now bears a negative charge instead. For basic amino acids, a basic coating induces a strong downregulation of their charge: this reduces repulsions with the protonated and positively charged monomers and, at the same time, decreases the overall positive net charge of the proteins.

So far, the results presented corresponded to single-protein solutions, containing either GFP or lysozyme. We gained insights on the effect of protein size, net charge, and intrinsic properties such as the isoelectric point on the protein adsorption process for each of them. Next, we consider solutions containing both proteins simultaneously.

### 3.3. Protein Mixtures: Competitive Adsorption

Figure 9 describes the behavior of binary GFP–lysozyme solutions adsorbing onto cylindrical NPs of 5 nm radius coated with either neutral, acidic, or basic polymer mixtures. As discussed for single-protein solutions, the electrostatic interactions and the variables that tune them (pH and ionic strength of the solution) are of paramount importance. However, in mixtures, competitive adsorption between proteins also comes into play, factoring in protein size and charge.

For the neutral coated NPs (Figure 9 left panel), lysozyme adsorption increases as the pH of the bulk solution approaches that of its isoelectric point (as discussed previously for the one-protein solutions). It is interesting to see that for the case of GFP in the protein mixture, though, the peak of adsorption is shifted towards higher pH values with respect with the one-protein solution (Figure 4, right panel). This could be rationalized considering the GFP–lysozyme electrostatic interactions as they approach the neutral surface. Given their pI values (6.33 and 10.9 for GFP and lysozyme, respectively), proteins have opposite charges, giving rise to attractive electrostatic interactions. These attractive forces can compensate the greater steric repulsions arising from GFP adsorption, given its bigger size (129 amino acids for lysozyme vs. 238 amino acids for GFP). The increase of GFP adsorption in this pH range occurs along with a decrease in the adsorption of lysozyme, as compared to lower or higher pH values. This shows the nontrivial balance between electrostatic and steric forces that arises when both proteins are in solution. As the pH approaches the pI of lysozyme, its charge goes down, increasing its adsorption, at the expense of GFP. Finally, for high pH values, both lysozyme and GFP are negatively charged, leading to a decrease in protein adsorption, as discussed previously for the one-protein solutions.

Now, when the coating of the NP is pH responsive, either acidic or basic, we observe that electrostatic protein–protein and protein–polymers interactions dominate the competitive adsorption process over the effect of steric interactions. Similar results have been described for proteins adsorbing onto pH-responsive hydrogels grafted on planar surfaces [72]. Conditions in which a protein in solution (either lysozyme or GFP) is oppositely charged to the polymer coating lead to an almost complete adsorption of that protein and depletion of the other protein in the mixture, irrespective of their sizes. Hence, we observe a complete dominance of lysozyme adsorption when the NP is grafted with acidic polymers, driven by both electrostatic attractions and steric interactions (similarly to the one-protein solution, Figure 4, center panel). For the basic coated NPs, we observe a more rich interplay between these forces. For high pH values (>9.0), polymers are mostly neutral, hence steric and protein–protein electrostatic interactions are the dominant forces. These forces favor lysozyme adsorption over GFP, given its smaller size, recovering the behavior observed for neutral coated NPs (as discussed above). Now, for pH values lower than 9.0, the polymer layer becomes positively charged, favoring the adsorption of the negatively charged GFP proteins at the expense of the adsorption of the smaller but positively charged lysozyme. However, once pH falls below its pI, $\Gamma_{GFP}$ decreases, driven by both electrostatic and steric repulsions with the polymer layer (similarly to the one-protein solution, Figure 4, right panel).

The above results highlight that the interplay of polymer and protein acid–base equilibria is key in determining the competitive protein adsorption process. Given this, it is interesting to analyze the effect of salt concentration, since it allows modulating the electrostatic interactions between all charged species in the system (proteins, titrable monomers, ions). Figure 10 describes the behavior of binary GFP–lysozyme solutions adsorbing onto cylindrical NPs of 5 nm radius coated with either neutral, acidic, or basic polymer mixtures for different salt concentrations.

Our calculations show that the effects of salt concentration depend on the type of polymer coating, the proteins present in the mixture, and the pH of the bulk solution. As discussed previously for lysozyme adsorbing onto surfaces grafted with neutral polymers, increasing salt concentration leads to an overall increase of $\Gamma_{lys}$ (Figure 3). The same trend is observed when having GFP in the mixture along with lysozyme (Figure 10 left upper panel), while the opposite is true for $\Gamma_{GFP}$ (Figure 10 left lower panel). Increasing salt concentration allows screening electrostatic repulsions between same-charged proteins, and since no other electrostatic forces are present for neutral coated NPs, this favors the adsorption of the smaller protein at the expense of the larger one in the mix. These results show that when having protein mixtures in solution, the competitive nature of the adsorption process makes the analysis protein-dependent.

When NPs are grafted with titrable monomers, electrostatic screening applies both to attractive and repulsive forces, again reinforcing the role of steric interactions. Increasing salt concentration decreases protein–polymers electrostatic attractions, and, with that, protein adsorption. The opposite is observed for protein–polymers electrostatic repulsions, leading to an increase in protein adsorption with increasing salt concentration. In this way, the resulting effect depends on the pH of the solution, the identity of the protein (its pI), and the type of coating (its pKa), since their interplay will determine the type of electrostatic interactions in the system. This has also been discussed on the basis of experimental measurements of the adsorption of lysozyme and β-lactoglobulin onto silica NPs for different salt concentrations [34]. Although GFP and β-lactoglobulin have very different sizes and structure, they have similar isoelectric points, making it possible to reutilize the discussion based on electrostatic interactions in the system. Changes in adsorption are also fueled by the competitive nature of the process when a protein mixture is present. Following this general analysis, our calculations show that for acidic coated NPs, increasing salt concentration drastically decreases lysozyme adsorption. However, given its size relative to GFP, even the highest salt concentration studied (0.1 M) results in almost no adsorption of GFP for the whole pH range. For NPs with a basic coating, increasing salt concentration screens GFP–polymer attractions, resulting in a decrease in its adsorption. Increase in lysozyme adsorption can be rationalized considering the screening effect in electrostatic protein–polymer repulsions and also the competition with GFP on adsorption: less GFP molecules on the surface leave more space for lysozyme molecules to adsorb.

To highlight even further the impact of charge regulation in the proteins upon adsorption, we analyzed how their overall charge changes once adsorbed onto surfaces modified with different coatings. Results for the average net charge of adsorbed lysozyme and GFP are collected in Figure 11. The computation of 〈Z${}_{prot}\rangle$ for each protein in the mixture was performed following an analogue procedure as described in Ref [62]. Our calculations show that adsorption onto a neutral polymer coating leads to minor modifications on the charge of the adsorbed protein as compared to its bulk value. Proteins average net charge are slightly lower than in bulk for the whole pH range, in line with our previous arguments of decreasing protein charge to minimize same charge protein–protein interactions. However, important changes in the isoelectric point of the proteins on the surface are observed when the polymer coating has chargeable monomers [72]. For both proteins adsorbed on the surface, our calculations show that acid coatings favor more positively charged species for the whole pH range (above the values for the bulk counterpart), while the opposite is true for basic coatings. In this way, electrostatic attractions between the adsorbing proteins and the polymer coating are maximized.

The local protein net charge also changes significantly upon adsorption, and is quite different than in solution. Appendix A shows how the charge of lysozyme and GFP changes as a function of the distance from a surface modified with the different coatings. Particularly interesting is to note that, when the two proteins are adsorbing together, regulation of charges is coupled between the two proteins. For example, we see that for the neutral NP at pH 7, GFP modifies its charge to more negative values in order to increase the attractive interactions with the positively adsorbed lysozyme. Now, for acidic coatings at this bulk solution pH, we observe charge inversion for GFP as it approaches the surface, going from ∼−4 in solution to ∼+6 on the surface to maximize attractive interaction with the negatively charged polymers.

Finally, we would like to address the role played by the adsorption energy of the proteins that we use in our calculations ($\epsilon_{ads,prot}$). Results and discussions so far correspond to a fixed value of $\epsilon_{ads,prot}=-50k_{B}T$ (124 KJ/mol for T = 298 K) for both lysozyme and GFP interactions with the bare surface. To explore the role played by this parameter, we performed calculations of protein mixtures in contact with cylindrical surfaces modified with neutral, acidic, or basic polymers, keeping the same adsorption energy for both proteins, increasing their values and finally changing the relative value of GFP adsorption energy with respect to the one for lysozyme, given its bigger size. Results are collected in Figure 12.

As to be expected, increasing the adsorption energy leads to a greater adsorption of both proteins (upper and middle row), irrespective of the type of polymer coating. It is interesting to see the increase in lysozyme adsorption for pH values < 5 for NPs modified with basic polymers. As we analyzed for the one-protein solutions, at these pH values both the lysozyme and the polymers are positively charged, leading to strong electrostatic repulsions, that, along with steric repulsions, result in almost no protein adsorption (Figure 4). However, increasing the adsorption energy starts to compete with these opposing forces, and for the values computed in Figure 12 top row, we see that the driving force for adsorption dominates, favoring the smaller protein in the mixture, hence leading to an increase in lysozyme adsorption. Finally, when increasing the relative value of GFP adsorption energy with respect to that for lysozyme, we observe a nontrivial rearrangement of forces, since increasing GFP adsorption energy competes with the steric repulsions derived from its bigger size and the protein–protein electrostatic repulsions.

For neutral coatings, we observe a change in behavior for pH values < 6. At those values, both proteins are positively charged, leading to electrostatic protein–protein repulsions, but the greater GFP adsorption energy favors its adsorption over lysozyme despite the protein size and charge. For the acidic coated NPs, we observe an increased adsorption of GFP for pH values between 3 and 5. In this range, there is a small fraction of negatively charged monomers (Appendix A) that interacts with the positively charged proteins. Note that in this range, GFP positive charge is larger than the positive charge of lysozyme (Figure 11). When the adsorption energy of the proteins is the same, the protein size and its derived steric repulsions dominate, favoring lysozyme adsorption. However, when increasing $\beta \epsilon_{ads}$ for GFP with respect to lysozyme, this can be contested, leading to an increased adsorption of the bigger protein. The same analysis can be carried out to rationalize the increased adsorption of GFP for low pH values, at the expense of lysozyme. Our calculations showed that changing the adsorption energy of the proteins in the mixture highlights the competitive nature of the adsorption process and the fact that the acting forces balance themselves in a nontrivial manner.

## 4. Conclusions

In this work, we investigated the nonspecific protein adsorption process onto surfaces of different morphology and curvature, modified with a mixture of polymers of different length and monomer type (neutral, acidic, basic). We resorted to a molecular theory that considers the size, shape, conformation, and charge of each molecular species in the system. In addition, very importantly, the theoretical framework allows to explicitly take into account the acid–base reactions of the titrable amino acids in the protein and the monomers in the grafted polymers. In this way, we are able to investigate the nontrivial balance between chemical equilibria and physical interactions for surface-engineered NPs in contact with solutions of different pH values, salt concentrations, and proteins. Specifically, the molecular model we used in our calculations allowed us to obtain protein adsorption isotherms as a function of relevant parameters (pH, salt concentration, surface properties), the local concentrations of mobile species, the spatially resolved molecular organization of all the species in the system, and the charge state of titrable group (amino acids or monomers), among others.

The adsorption from single-protein solutions of lysozyme and GFP onto planar, cylindrical, and spherical surfaces modified with a mixture of neutral polymers showed that even though surface curvature modulates the adsorption process and the molecular organization in space, the dominant interactions in the systems are electrostatic in nature, besides steric repulsions and surface–protein attractions. Hence, pH and ionic strength (in the millimolar range) are the key players in the process. Regarding pH, protein adsorption increases as we span from low to high pH values, and it is maximum when the pH of the bulk solution is similar to the isoelectric point of the protein. In systems of neutral NPs, the only electrostatic interactions are the repulsions between same-charge adsorbing protein molecules: less charge, less repulsions, more adsorption. Along the same line, increasing salt concentration leads to a decrease in electrostatic repulsions and, with that, to an increase in the amount of adsorbed protein. However, the magnitude of this effect depends on the pH. Finally, for NPs coated with neutral polymers, increasing the amount of total monomers on the surface (either by increasing the surface density of polymers or increasing the fraction of long polymers in the surface mixture) hinders protein adsorption by presenting a steric barrier towards approaching protein molecules.

Playing with the nature of the long polymer on the surface (making it acidic or basic) gave us an opportunity to study the effects of electrostatic protein–polymers interactions. In these systems, the adsorption process becomes protein-specific (through its size and charge) and it is dominated by the protein’s pI value relative to the pKa of the surface polymer. The pH of the contacting solution gives us a proxy to manipulating the molecular interactions in the system, going from electrostatic repulsions to attractions. This means that the same polymer coating can act as preventing or promoting protein adsorption depending on the conditions of the solution, which make these smart systems sensitive and responsive to the environment. Now, the effect of the salt concentration is not trivial, as salt ions act by screening the charges in the system. For conditions in which protein–polymer electrostatic interactions are attractive, an increase in salt concentration would lead to a decrease in protein adsorption, while the opposite is true if the dominant interactions are repulsive. Similarly, increasing polymer density on the surface does not necessary translate into a decrease in protein adsorption: it depends on the nature of the electrostatic interactions between proteins and polymers.

Our calculations allowed us to explore the electrostatic coupling of protonation and deprotonation reactions of both titrable amino acids in proteins and chargeable monomers on the NPs’ surface. Proteins are ampholites with titrable amino acids capable of acquiring positive or negative charges. Charge regulation is a balance between upregulating and downregulating degree of charge of these titrable groups in order to minimize electrostatic repulsions and maximize attractions between charged species in the system. A similar charge regulation effect was observed for grafted basic or acidic polymers.

Finally, having a mixture of proteins in solution leads to a competitive adsorption process that can be modulated by the conditions of the solution (pH and salt concentration). Moreover, given the nontrivial nature of this process, one must know the proteins in solution and their physicochemical properties (pI and size) in order to design a system or determine the conditions that would favor or disfavor adsorption.

In order to advance on the results obtained thus far and study multilayer protein adsorption, we need to explore the role of electrostatics and hydrophobic interactions on the adsorption process [5], also considering the possibility of weakly attracting polymers to proteins. These theoretical calculations will be combined with experiments on protein adsorption, to advance previous results from our group on prevention of protein adsorption by PEG [76].

Future lines of work derive from the assumptions of the model employed in this work. Expanding our calculations from 1D to 3D systems (that is, without considering lateral homogeneity as we have done in this work) would allow us to study clustering of proteins, the role of surface properties, and whether this process is surface-mediated [77]. We would like to explore how protein changes upon adsorption can alter the process. Particularly, we are interested in studying the influence of adsorption on the proteins orientation on the surface, and how it depends on the design parameters of the engineered surface and the conditions of the solution (pH, salt concentration) with the intention of comparison to available experimental results of similar systems [27] and those that will be provided by our experimental collaborators. These studies would also allow us to contribute to the current discussion on proteins adsorbing on the “wrong side” of their isoelectric point [78,79].

## Figures and Tables

**Figure 1 polymers-14-00739-f001:**
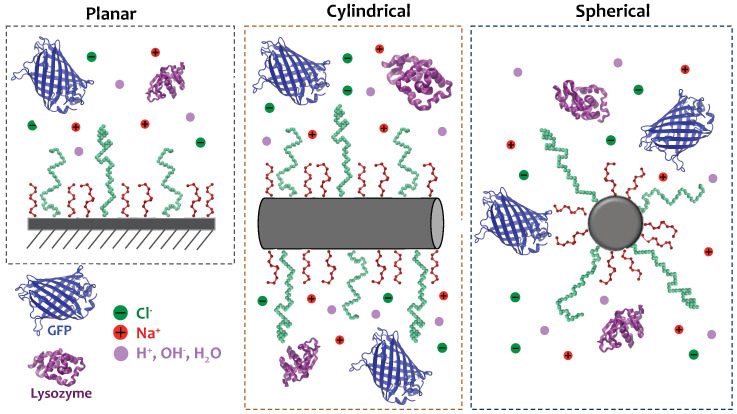
Schematic representation of the systems modeled. Surfaces of different morphology and curvature are end-tethered with a mixture of short and long hydrophilic polymers. The long polymers can be neutral, acidic, or basic. Surface composition (% of long polymers in the surface mixture) and total polymer surface density are controlled. The modified surfaces are in contact with an aqueous solution containing a mixture of proteins and ions. The pH, salt, and protein concentrations of the bulk solution are controlled.

**Figure 2 polymers-14-00739-f002:**
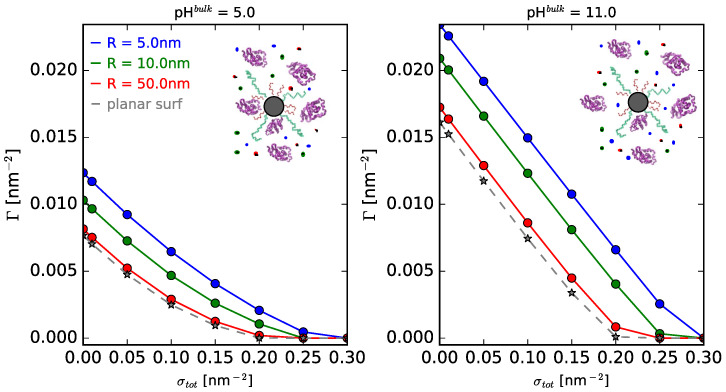
Adsorption isotherms of lysozyme onto spherical NPs of different sizes as a function of total polymer surface density ($\sigma_{tot}$) for two given pH values. (**left panel**) pH = 5.0 and (**right panel**) pH = 11.0. Long polymers are neutral and the surface composition is fixed, x${}_{l}^{surf}$ = 0.5. c${}_{salt}$ = 1 mM; c${}_{lys}$ = 10${}^{-4}$ M. The radii of the spherical NPs are indicated in the legend, as well as the limiting case for a planar surface.

**Figure 3 polymers-14-00739-f003:**
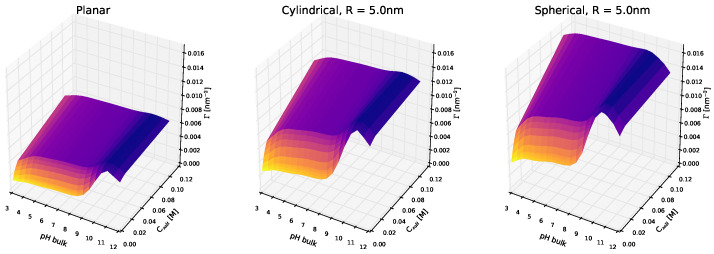
Effect of pH and salt concentration on the adsorption of lysozyme onto planar, cylindrical, and spherical NPs. For the curved systems, R = 5 nm. Long polymers are neutral, $\sigma_{tot}$ = 0.1 nm${}^{-2}$, and x${}_{l}^{surf}$ = 0.5. The bulk protein concentration is fixed, c${}_{lys}$ = 10${}^{-4}$ M. Note that all panels are in the same scale.

**Figure 4 polymers-14-00739-f004:**
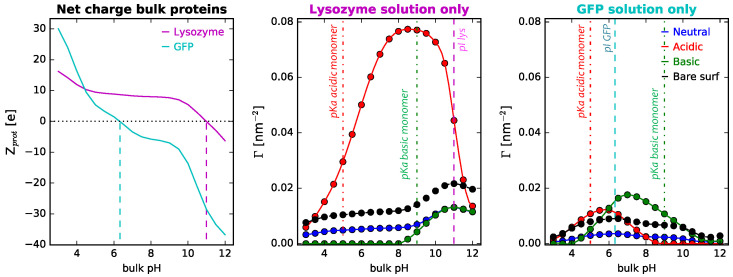
(**left panel**) Computed net charge of lysozyme and GFP in dilute solution as a function of bulk pH. Dashed lines indicate the isoelectric point of each protein (10.99 and 6.33 for lysozyme and GFP, respectively). (**center** and **right panels**) Protein adsorption onto cylindrical NPs with R = 5 nm. Long polymers are neutral, acidic, or basic, as indicated in the legend. Total polymer surface density is fixed, $\sigma_{tot}$ = 0.1 nm${}^{-2}$, and the surface composition is x${}_{l}^{surf}$ = 0.5. The bulk solution conditions are c${}_{salt}$ = 1 mM and c${}_{lys}$ = 10${}^{-4}$ M. Dashed lines correspond to the computed isoelectric points of the proteins, while the line-dot lines correspond to the pKa of the acidic and basic monomers (pKa${}^{acid}$ = 5.0; pKa${}^{basic}$ = 9.0).

**Figure 5 polymers-14-00739-f005:**
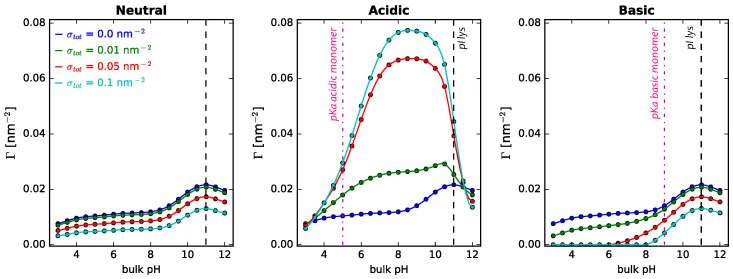
Effect of surface details on lysozyme adsorption onto cylindrical NPs, R = 5 nm. Long polymers are neutral, acidic, or basic, as indicated in the the header of each panel. The composition of the surface polymer mixture is fixed, x${}_{l}^{surf}$ = 0.5, while the total polymer surface density is varied, as indicated in the legend. The bulk solution conditions are c${}_{salt}$ = 1 mM and c${}_{lys}$ = 10${}^{-4}$ M. The dashed black lines correspond to the isoelectric point of lysozyme in dilute solution (pI = 10.99), while the magenta line-dot lines in the central and right panels correspond to the pKa of the acidic and basic monomer, respectively (pKa${}^{acid}$ = 5.0; pKa${}^{basic}$ = 9.0). The corresponding plots for GFP adsorption can be found in the Appendix A.

**Figure 6 polymers-14-00739-f006:**
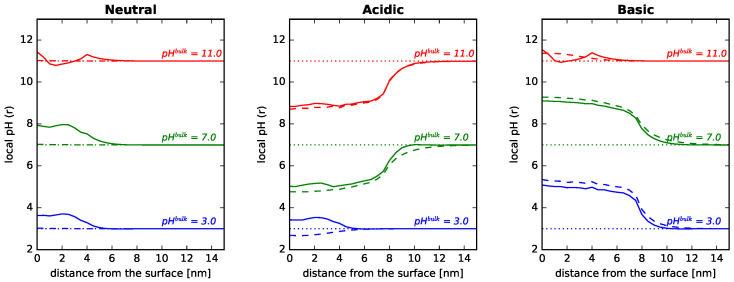
Local pH as a function of the distance to the surface for the adsorption of lysozyme onto cylindrical NPs, R = 5 nm. Long polymers are neutral, acidic, or basic, as indicated in the the header of each panel. Surface details are fixed, $\sigma_{tot}$ = 0.1 nm${}^{-2}$ and x${}_{l}^{surf}$ = 0.5. The bulk solution conditions are c${}_{salt}$ = 1 mM and c${}_{lys}$ = 10${}^{-4}$ M, while the pH was changed, as indicated in the legends. Note that the amount of adsorbed protein is different for each bulk pH (see Figure 4). Dotted lines correspond to the bulk pH value, while dashed lines correspond to the local pH without any protein in solution. The corresponding plots for GFP adsorption can be found in the Appendix A.

**Figure 7 polymers-14-00739-f007:**
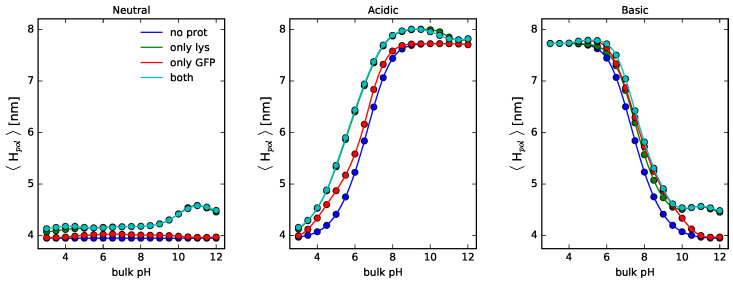
Average polymer height as a function of bulk pH for cylindrical NPs, R = 5 nm. Long polymers are neutral, acidic, or basic, as indicated in the the header of each panel. Surface details are fixed, $\sigma_{tot}$ = 0.1 nm${}^{-2}$ and x${}_{l}^{surf}$ = 0.5. c${}_{salt}$ = 1 mM. Lines correspond to the polymer layer in contact with a solution containing no proteins, lysozyme only, GFP only, or a binary mixture of lysozyme and GFP, as indicated in the legend.

**Figure 8 polymers-14-00739-f008:**
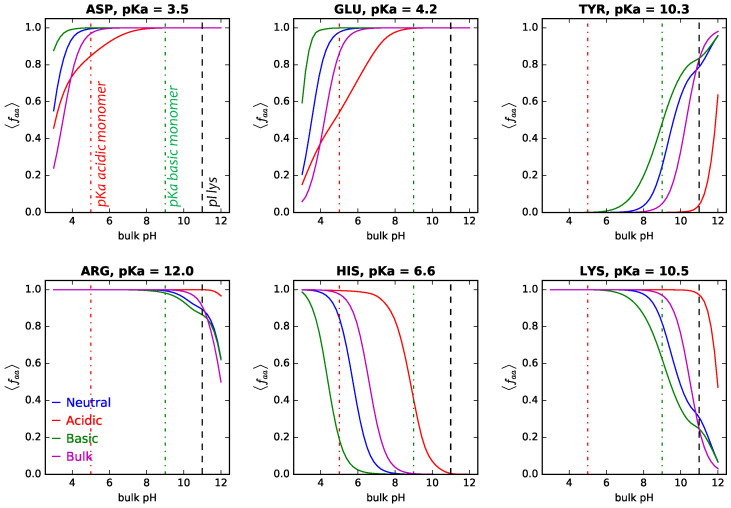
Average of the fraction of deprotonated (acidic, upper panels) or protonated (basic, lower panels) amino acids of adsorbed lysozyme onto cylindrical NPs (R = 5 nm) as a function of bulk pH. The amino acid name, type, and pKa are indicated in the header of each panel. Surface details are fixed, $\sigma_{tot}$ = 0.1 nm${}^{-2}$ and x${}_{l}^{surf}$ = 0.5. The bulk solution conditions are c${}_{salt}$ = 1 mM and c${}_{lys}$ = 10${}^{-4}$ M. Blue, red, and green full lines correspond to NPs grafted with neutral, acidic, and basic polymers, as indicated in the legend. Magenta lines correspond to the amino acid in bulk solution. Vertical lines correspond to the pI of lysozyme and the pKa of the acidic and basic monomers.

**Figure 9 polymers-14-00739-f009:**
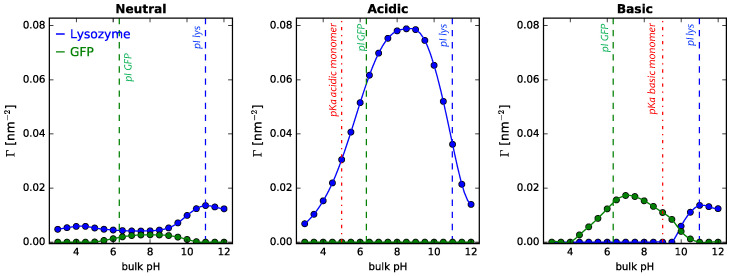
Adsorption of lysozyme and GFP from binary mixtures in solution onto cylindrical NPs, R = 5 nm, as a function of bulk pH. Long polymers are neutral, acidic, or basic, as indicated in the the header of each panel. Surface details are fixed, $\sigma_{tot}$ = 0.1 nm${}^{-2}$ and x${}_{l}^{surf}$ = 0.5. The bulk solution conditions are c${}_{salt}$ = 1 mM and c${}_{lys}$ = c${}_{GFP}$ = 10${}^{-4}$ M. The dashed lines correspond to the isoelectric points of lysozyme and GFP in dilute solution, while the line-dot lines in the central and right panels correspond to the pKa of the acidic and basic monomer, respectively.

**Figure 10 polymers-14-00739-f010:**
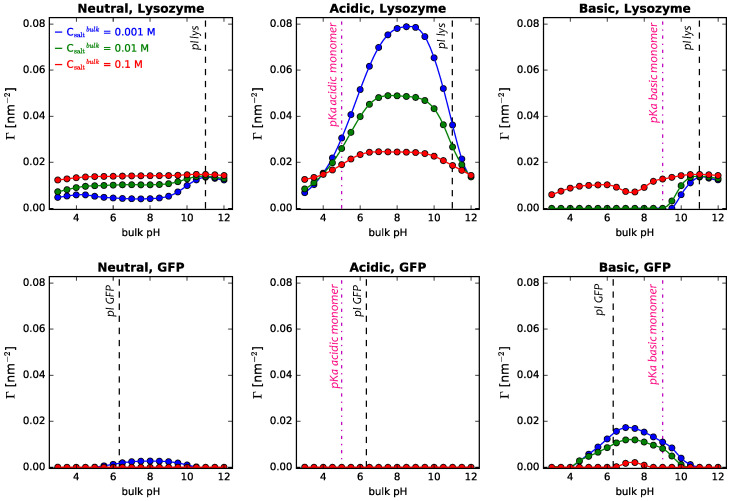
Salt concentration effect on the adsorption of lysozyme (**upper panels**) and GFP (**lower panels**) form binary mixtures in solution onto cylindrical NPs, R = 5 nm, as a function of bulk pH. Long polymers are neutral, acidic, or basic, as indicated in the the header of each panel. Surface details are fixed, $\sigma_{tot}$ = 0.1 nm${}^{-2}$ and x${}_{l}^{surf}$ = 0.5. The bulk solution conditions are c${}_{lys}$ = c${}_{GFP}$ = 10${}^{-4}$ M, and the salt concentration is varied, as indicated in the legend. The dashed black lines correspond to the isoelectric points of lysozyme and GFP in dilute solution, while the line-dot lines correspond to the pKa of the acidic and basic monomers (central and right panels, respectively).

**Figure 11 polymers-14-00739-f011:**
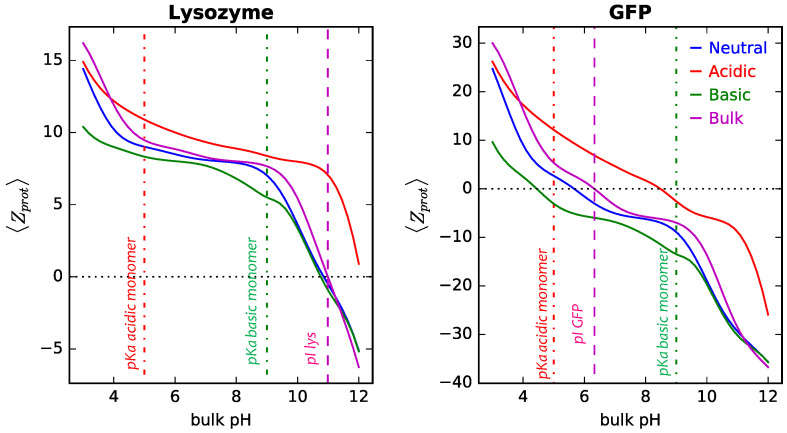
Average net charge of adsorbed lysozyme (**left panel**) and GFP (**right panel**) form binary mixtures in solution onto cylindrical NPs, R = 5 nm, as a function of bulk pH. Long polymers are neutral, acidic, or basic, as indicated in the legend. The magenta full lines correspond to the charge of the protein in bulk solution. Surface details are fixed, $\sigma_{tot}$ = 0.1 nm${}^{-2}$ and x${}_{l}^{surf}$ = 0.5. The bulk solution conditions are c${}_{lys}$ = c${}_{GFP}$ = 10${}^{-4}$ M and c${}_{salt}$ = 1 mM. The dashed vertical lines correspond to the isoelectric points of lysozyme and GFP in dilute solution, while the line-dot lines correspond to the pKa of the acidic and basic monomer, respectively. The dotted horizontal lines in each panel give the conditions for 〈Z${}_{prot}\rangle$ = 0.

**Figure 12 polymers-14-00739-f012:**
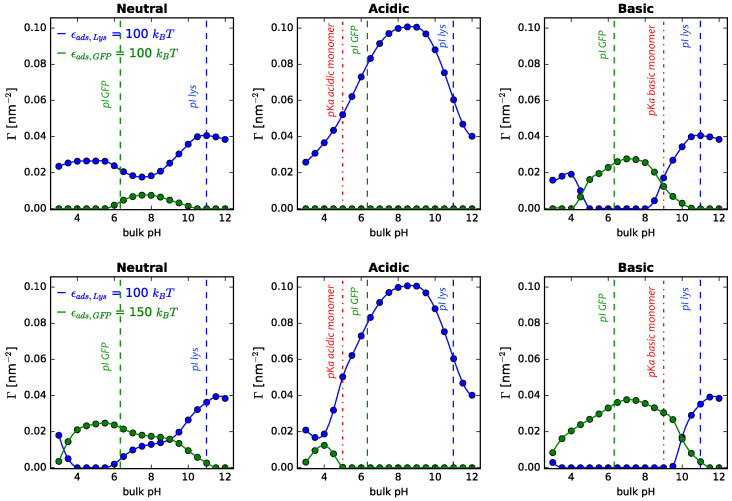
Adsorption of lysozyme and GFP from binary mixtures in solution onto cylindrical NPs, R = 5 nm, as a function of bulk pH for different adsorption energies. Each row correspond to a different adsorption energy (upper row: $\epsilon_{ads,Lys}$ = 100 $k_{B}T$, $\epsilon_{ads,GFP}$ = 100 $k_{B}T$; lower row: $\epsilon_{ads,Lys}$ = 100 $k_{B}T$, $\epsilon_{ads,GFP}$ = 150 $k_{B}T$, as indicated in the legend). Long polymers are neutral, acidic, or basic, as indicated in the the header of each panel. Surface details are fixed, $\sigma_{tot}$ = 0.1 nm${}^{-2}$ and x${}_{l}^{surf}$ = 0.5. The bulk solution conditions are c${}_{salt}$ = 1 mM and c${}_{lys}$ = c${}_{GFP}$ = 10${}^{-4}$ M. The dotted lines correspond to the isoelectric points of lysozyme and GFP in dilute solution, while the line-dot lines in the central and right panels correspond to the pKa of the acidic and basic monomer, respectively (pKa${}^{acid}$ = 5.0, pKa${}^{basic}$ = 9.0).

## Data Availability

Not applicable.

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
