# Peer review of "Proteins Adsorbing onto Surface-Modified Nanoparticles: Effect of Surface Curvature, pH, and the Interplay of Polymers and Proteins Acid–Base Equilibrium"

_polymers, 2022, doi:10.3390/polym14040739_

Round 1

Reviewer 1 Report

Manuscript titled as “Proteins Adsorbing onto Surface Modified Nanoparticles: Effect of Surface Curvature, pH and the Interplay of Polymers and Proteins acid-base Equilibrium” describes new and interesting theoretical calculations and can be published in Polymers mdpi after major revision.

First and main remark, Supporting Information that was mentioned repeatedly in manuscript is absent in access.

Unfortunately, theoretical calculations were not confirmed by experimental results. I believe that these assumptions are very valuable and probably may be used for prediction of the proteins behavior but chosen system is too complicated to take into account all possible factors.

Information about acid-base character of the hypotetical polymers is too limited, as known are strong polyacids/polybase and weak polyacids/polybase.  I also did not find any information about molecular weight hypothetical polymers. Appropriate discussion should be provided.

I would like to recommend to cite following references where similar results were presented.

https://doi.org/10.1021/acs.biomac.9b00030

https://doi.org/10.1021/acsami.7b00136

Author Response

Comments and Suggestions for Authors #1

Reviewer’s Comment #1: Manuscript titled as “Proteins Adsorbing onto Surface Modified Nanoparticles: Effect of Surface Curvature, pH and the Interplay of Polymers and Proteins acid-base Equilibrium” describes new and interesting theoretical calculations and can be published in Polymers mdpi after major revision.

We thank the reviewers for their time and comments on our manuscript. The revisions needed are incorporated in the re-submitted version and are answered below.

Reviewer’s Comment #2: First and main remark, Supporting Information that was mentioned repeatedly in manuscript is absent in access.

We apologize for the mistake. Inadvertently, the Electronic Supporting Information (ESI) was not uploaded in pdf version along the main manuscript, although it was added in the zip file with the rest of the Latex files. It has now been added.

Reviewer’s Comment #3: Unfortunately, theoretical calculations were not confirmed by experimental results. I believe that these assumptions are very valuable and probably may be used for prediction of the proteins behavior but chosen system is too complicated to take into account all possible factors.

We appreciate the reviewer’s comments and insights. Indeed, there are no accompanying experimental results in this work, which is based on theoretical modeling of the protein adsorption onto NPs. However, references to experimental work are present along the manuscript, in line with the results provided by our calculations. The reviewer is right in pointing out the value of combined experimental and theoretical work, and in that line future work will include combined theoretical calculations combined with experiments on protein adsorption onto modified NPs provided by our experimental collaborators, following previous work in which planar surfaces were modified with PEG to prevent protein adsorption (lines 759-761). However, previous applications of the molecular theory used in this work to explore protein adsorption onto planar surfaces (and polymer adsorption onto curved NPs) give us confidence in that our current work could provide new insights, expanding those works into curved NPs.

Regarding the complexity of the system, the reviewer is correct in pointing out that the systems explored in the manuscript pose a challenge for synthesis and preparation of the materials in the lab and their further characterization in contact with solutions containing proteins. However, we believe that this is an opportunity for theory to explore parameters in a systematic and exhaustive way, which might be difficult to do experimentally. In this way, our theoretical calculations may contribute to a fundamental understanding of the physical-chemistry of the system and on how the protein adsorption depends on the design parameters of the modified NPs. This is discussed in the introduction, when summarizing previous results (lines 79-96) and pointing out what we believe to be the knowledge gap we try to address with our current work.

Reviewer’s Comment #4: Information about acid-base character of the hypothetical polymers is too limited, as known are strong polyacids/polybase and weak polyacids/polybase. I also did not find any information about molecular weight hypothetical polymers. Appropriate discussion should be provided.

The reviewer’s comment is on spot and it is related to the missing file for the ESI. Indeed, in it there is the section “Molecular models: Grafted Polymers” describing the polymer model and the details on the polymer mixture decorating the surface of the NPs. To further expand the characterization of the modeled polymers, we have added the following text: “Long polymers have 45 monomers, while short ones have 10. The choice of polymer size follows the commonly commercialized PEG products with molecular weights of 2000 and 500, respectively. The monomer volume was considered the same in all types of polymers and equal to that of PEG, as done in previous calculations [9]”. The added text is marked in blue in the revised version of the ESI (lines 288-292 in ESI).

Reviewer’s Comment #5: I would like to recommend to cite following references where similar results were presented:

https://doi.org/10.1021/acs.biomac.9b00030 (Temperature-controlled orientation of proteins on temperature responsive grafted polymer brushes. Poly(butyl methacrylate) versus poly(butyl acrylate): Morphology, wetting and protein adsorption)

https://doi.org/10.1021/acsami.7b00136 (Temperature-Controlled Three-Stage Switching of Wetting, Morphology, and Protein Adsorption)

We thank the reviewer for the references. In those works, temperature-responsive coatings onto planar glass substrates are studied, specifically the impact of temperature on protein adsorption onto them. Bovine serum albumin (BSA) and anti-IgG were used in the experiments. The authors found a strong temperature dependence of protein adsorption, increasing with temperature, while also critically affecting the orientation of the proteins on the surface. The influence of pH on the thermo-sensitivity of modified surfaces was also studied. Given the relevance of the work performed, the references were added in the Introduction when summarizing the strategies to prevent and control protein adsorption on planar systems. The added text is marked in blue in the revised version (line 50).

The first reference is also mentioned in the Conclusions, when discussing future work on protein orientation upon adsorption onto modified surfaces. The added text is marked in blue in the revised version (lines 768-771).

Reviewer 2 Report

Dear Authors, manuscript Proteins Adsorbing onto Surface Modified Nanoparticles: Effect of Surface Curvature, pH and the Interplay of Polymers and Proteins acid-base Equilibrium is very interesting. The presented results and very good and deep discussion, clearly show the necessity of consideration of the modification of NP on a molecular level. This is especially important when are available methods for direct protein design and also polymer-based carriers for developing e.g new biosensors or effective therapeutic proteins releasing systems. The manuscript is well written and only small corrections are necessary.

Comments;

  1. 43-51; Authors rightly highlighted that NPs can be modified by various polymers for reduction nonspecific protein adsorption. However, what exactly means nonspecific it is not desired adsorption or not correctly oriented especially in the case of enzymes? This phrase strongly suggests a reduction of the possibility of modification NPs surface by polymers.
  2. Line 51; ….. more proteins that neutral or hydrophilic ones; rather should be; more proteins than neutral or hydrophilic ones
  3. Line 113-123; The aim of the study should be possibly short and contain a clear working hypothesis.
  4. Line 233; Authors wrote; This follows from the fact that protein-protein interactions are of repulsive nature…. The phrase should be more precise and reference should be also added. What with the oligomeric proteins where the correct conformation of oligomer is the resultant of “attraction and repulsion” ?

Author Response

Comments and Suggestions for Authors #2

Reviewer’s Comment #1: Dear Authors, manuscript “Proteins Adsorbing onto Surface Modified Nanoparticles: Effect of Surface Curvature, pH and the Interplay of Polymers and Proteins acid-base Equilibrium” is very interesting. The presented results and very good and deep discussion, clearly show the necessity of consideration of the modification of NP on a molecular level. This is especially important when are available methods for direct protein design and also polymer-based carriers for developing e.g new biosensors or effective therapeutic proteins releasing systems. The manuscript is well written and only small corrections are necessary.

We are grateful for the time and dedication of the reviewer towards our manuscript. Below we address every comment separately.

Reviewer’s Comment #2: 43-51; Authors rightly highlighted that NPs can be modified by various polymers for reduction nonspecific protein adsorption. However, what exactly means nonspecific it is not desired adsorption or not correctly oriented especially in the case of enzymes? This phrase strongly suggests a reduction of the possibility of modification NPs surface by polymers.

Referred original text: “To control the nonspecific protein adsorption, the surface of the NPs are commonly passivated by functionalization with biocompatible polymers, [16] such us poly(ethylene)glycol (PEG), [17] self-assembled monolayers (SAMS), [18, 19] zwitterions, [20, 21] polysacharides, [22] peptoids. [23-25] The polymeric layer provides a repulsive steric barrier to proteins, limiting their adsorption. [26, 27] The efficiency of this anti-fouling strategy depends on the molecular weight of the polymer, its surface density. [28] It has been found that hydrophobic or charged NPs tend to adsorb more proteins than neutral or hydrophilic ones. [13, 29, 30]”

We thank the reviewer for bringing attention to this issue. By nonspecific protein adsorption we refer to adsorption driven by interaction with the surface without any engineered modification to it that would render it more specific, e.g., modifying the surface with ligands that would bind selectively to a group or moiety of the protein (or enzyme). In that line, the major factors determining non-specific protein adsorption are protein-substrate adsorption energy, their electrostatic interactions as well as van-der-Waals interactions, and hydrogen bonds. In this work, we explore the effects of surface-proteins adsorption energy and the effect of pH-dependent electrostatic interactions when modifying the NP with weak polyelectrolytes. To further clarify this, we have modified the original text, and added the following text: “This adsorption is non-specific in nature, meaning it does not follow a molecular recognition interaction but rather protein-surface attractions, electrostatic and van der Waals interactions.” The change is marked in blue in the revised version (lines 39-42).

Reviewer’s Comment #3: Line 51; ….. more proteins that neutral or hydrophilic ones; rather should be; more proteins than neutral or hydrophilic ones

Referred original text: “It has been found that hydrophobic or charged NPs tend to adsorb more proteins that neutral or hydrophilic ones.”

We appreciate the reviewers careful reading and spotting the typo. We had made the corresponding correction and the text now reads: “It has been found that hydrophobic or charged NPs tend to adsorb more proteins than neutral or hydrophilic ones.” The change is marked in blue in the revised version (lines 53-54).

Reviewer’s Comment #4: Line 113-123; The aim of the study should be possibly short and contain a clear working hypothesis.

Referred original text: “The paper is organized as follows: in the next section we provide a description of the theoretical approach and molecular models for proteins and polymers employed in our calculations. We start presenting the results for protein adsorption from one protein solutions onto NPs of different curvature and morphology modified with neutral coatings. Analysis of the effects of the solution's pH and salt concentration in these neutral NPs is followed by the study of the interplay between the acid-base equilibrium of polymers and proteins for NPs functionalized with weak polyelectrolytes. Competitive adsorption from binary protein mixtures is then discussed. Finally, we conclude by presenting the main points of this work, tying together the key factors that govern the process of protein adsorption onto modified NPs, and by proposing highlights for future lines of work.”

We thank the reviewer for their comment and suggestion. We believe the proposed addition would be better suited in the previous paragraph to the cited lines. In it, we have added the following text: “Our working hypothesis is that the conditions of the solution (pH and ionic strength), the details of the engineered surface (type of polymers, their surface density), and the coupled charge regulation between proteins and grafted polymers (when acidic or basic), modulated by the surface curvature, determine the protein adsorption process.” The change is marked in blue in the revised version (lines 103-107).

Reviewer’s Comment #5: Line 233; Authors wrote; This follows from the fact that protein-protein interactions are of repulsive nature…. The phrase should be more precise and reference should be also added. What with the oligomeric proteins where the correct conformation of oligomer is the resultant of “attraction and repulsion” ?

Referred original text: “This follows from the fact that protein-protein interactions are of repulsive nature (both electrostatic and steric), since no Van der Waals interactions were included in the current model.”

The reviewer raises an interesting point. However, the referred sentence corresponds to the results obtained from our calculations, with the limitations of the theoretical and molecular model we employ. We realize that the wording of the sentence makes it unclear, so we have modified it in the following way: “This follows from the fact that in our current model protein-protein interactions are of mainly repulsive (both electrostatic and steric), since no Van der Waals interactions were included.” The change is marked in blue in the revised version (lines 239-241).

Round 2

Reviewer 1 Report

Paper can be accepted in present form.